# MicroRNA-34/449 controls mitotic spindle orientation during mammalian cortex development

Juan Pablo Fededa[1,*], Christopher Esk[1], Beata Mierzwa[1], Rugile Stanyte[1], Shuiqiao Yuan[2], Huili Zheng[2], Klaus Ebnet[3], Wei Yan[2], Juergen A Knoblich[1] & Daniel W Gerlich[1,**]

## Abstract

Correct orientation of the mitotic spindle determines the plane of cellular cleavage and is crucial for organ development. In the developing cerebral cortex, spindle orientation defects result in severe neurodevelopmental disorders, but the precise mechanisms that control this important event are not fully understood. Here, we use a combination of high-content screening and mouse genetics to identify the miR-34/449 family as key regulators of mitotic spindle orientation in the developing cerebral cortex. By screening through all cortically expressed miRNAs in HeLa cells, we show that several members of the miR-34/449 family control mitotic duration and spindle rotation. Analysis of miR-34/449 knockout (KO) mouse embryos demonstrates significant spindle misorientation phenotypes in cortical progenitors, resulting in an excess of radial glia cells at the expense of intermediate progenitors and a significant delay in neurogenesis. We identify the junction adhesion molecule-A (JAM-A) as a key target for miR-34/449 in the developing cortex that might be responsible for those defects. Our data indicate that miRNA-dependent regulation of mitotic spindle orientation is crucial for cell fate specification during mammalian neurogenesis.

**Keywords** cortex development; miR-449; neurogenesis; radial glia; spindle orientation

**Subject Categories** Cell Adhesion, Polarity & Cytoskeleton; Neuroscience; RNA Biology

The EMBO Journal (2016) 35: 2386–2398

## Introduction

MicroRNAs (miRNAs) are small noncoding RNAs, which regulate gene expression posttranscriptionally by influencing mRNA stability and/or translation (Ambros, 2004; Du & Zamore, 2005; Kim, 2005). miRNAs are required for brain development in mammals, and growing evidence suggests that they play key roles during cortical neurogenesis (Bian & Sun, 2011). General inhibition of miRNA biogenesis through the conditional deletion of Dicer during early mouse development suppressed the generation of cortical neurons, but not the maintenance of the neural progenitor pool (De Pietri Tonelli et al, 2008; Kawase-Koga et al, 2009; Nowakowski et al, 2011). Since cortical neurons are generated from radial glial cells, these findings suggest that radial glial cell specification might be under the control of miRNAs. Two miRNA families, miR-17~92 and miR-7, were shown to promote the self-renewal of radial glial cells (Bian et al, 2013; Nowakowski et al, 2013; Fei et al, 2014) and the survival and differentiation of neural progenitors (Pollock et al, 2014), respectively. The miR-34b/c and miR-449 cluster encodes for six miRNAs, which are also relevant for brain development, as their deletion in mice reduced brain size and caused defects in basal forebrain structures (Wu et al, 2014). Yet, how these miRNAs contribute to specific stages of cortical development has remained unclear.

Mammalian corticogenesis relies on a precise regulation between the generation of neural progenitors and their differentiation (Gotz & Huttner, 2005). This balance is necessary to produce the large number of neurons that shape the multilayered cortical structure. At the onset of cortical neurogenesis, the expansion of neural progenitors by symmetric cell divisions declines and cortical neurons are generated through a series of asymmetric mitoses (Gotz & Huttner, 2005; Lancaster & Knoblich, 2012; Williams & Fuchs, 2013). After their amplification, neuroepithelial progenitors convert into radial glial progenitors, which migrate their nuclei to divide at the apical surface of the ventricular zone (Taverna & Huttner, 2010). Radial glial progenitors then generate cortical neurons by two different modes of asymmetric cell division (Malatesta et al, 2000; Anthony et al, 2004). Direct neurogenesis generates one radial glial progenitor and one neuron, whereas indirect neurogenesis generates one radial glial progenitor and an intermediate progenitor cell (also called basal progenitor), which ultimately generates a pair of neurons (Noctor et al, 2001, 2004; Calegari et al, 2002; Haubensak et al, 2004; Miyata et al, 2004). The degree of asymmetry in cell divisions of radial glial progenitors is influenced by mitotic spindle orientation (Lancaster & Knoblich, 2012; Williams & Fuchs, 2013). During early corticogenesis, parallel orientation of the spindle

1  Institute of Molecular Biotechnology of the Austrian Academy of Sciences (IMBA), Vienna Biocenter (VBC), Vienna, Austria
2  Department of Physiology and Cell Biology, University of Nevada School of Medicine, Reno, NV, USA
3  Institute-associated Research Group "Cell Adhesion and Cell Polarity", Institute of Medical Biochemistry, ZMBE, Münster, Germany
   *Corresponding author. Tel: +43 1 7904 44762; E-mail: jpfededa@gmail.com
   **Corresponding author. Tel: +43 1 7904 44760; E-mail: daniel.gerlich@imba.oeaw.ac.at

   

relative to the ventricular plate is essential for maintaining symmetric divisions. During the subsequent neurogenic phase, oblique and vertical spindle orientation shifts the balance toward indirect neurogenesis (Wynshaw-Boris, 2013; Xie *et al*, 2013).

Several molecular components and their mechanisms of regulating spindle orientation during cortical neurogenesis have been established through prior work (Postiglione *et al*, 2011; Xie *et al*, 2013). However, little is known regarding the contribution of miRNAs and posttranscriptional gene regulation to proper spindle orientation.

To elucidate the function of specific miRNAs during neural progenitor cell division in cortical development, we screened for candidate regulators of mitosis and subsequently characterized their function in mutant mice. We identified the miR-34/449 family as a critical factor for correct neural progenitor spindle orientation. MiR-34/449 KO mouse embryos had radial glial cells with misoriented spindles, resulting in reduced generation of intermediate progenitors via indirect neurogenesis and smaller cortices. We discovered that miR-449 targets the spindle regulator JAM-A for posttranscriptional repression *in vivo*. Thus, our findings show that the miR-34/449 family controls spindle orientation and the division of neural progenitors during development, with profound implications in mammalian cortical neurogenesis.

# Results

## A screen to identify candidate miRNAs involved in cortical cell division

To identify microRNAs (miRNAs) regulating spindle orientation during cortical neurogenesis, we developed an *in vitro* screening approach to search for candidates that affect cell division. We assayed mitotic duration in a HeLa cell line stably expressing a chromatin marker (histone 2B fused to a red fluorescent protein; H2B–mCherry) and a nuclear import substrate (importin-β-binding domain of importin-α fused to monomeric enhanced green fluorescent protein; IBB–eGFP) using live-cell microscopy (Schmitz *et al*, 2010). We individually transfected cells with a library of 135 miRNA-mimicking oligomers (miRNA mimics) representing all miRNAs expressed during mammalian corticogenesis (Yao *et al*, 2012) and, after 48-h incubation, recorded 24-h time-lapse movies for each miRNA mimic (Fig 1A). Mitotic duration was automatically determined for each dividing cell using supervised machine learning (Held *et al*, 2010). A miR-449b mimic caused the most substantial mitotic delay in this screen. MiR-449a and miR-34c mimics, which belong to the same miRNA family (Kozomara & Griffiths-Jones, 2014), also ranked within the top candidate hits with mitotic delays (Fig 1B and C, Appendix Table S1).

To further characterize the cellular defects associated with the mitotic delays, we investigated spindle rotation dynamics by confocal time-lapse microscopy using a HeLa cell line expressing H2B–mCherry and α-tubulin fused to monomeric enhanced green fluorescent protein (meGFP–α-tubulin; Steigemann *et al*, 2009). Transfection of the miR449a mimic caused excessive spindle rotation in cells that were delayed in mitosis, but also in cells that progressed to anaphase with normal timing (Fig 1D and E). This suggests that excessive mitotic spindle rotation is a direct

consequence of miR449a mimic transfection and not necessarily linked to prolonged mitosis. Thus, miRNA-34/449 family members might be involved in mitotic spindle orientation during brain cortex development.

## miR-34/449 family is expressed in the radial glial cell niche during cortical neurogenesis

The six members of the miRNA-34/449 family are expressed from three different loci (Fig EV1A). To investigate the role of miR-34/449 family during cortical development, we first determined whether the different members of the family are expressed in mice during the onset of cortical neurogenesis, at embryonic day 14 (E14). Laser capture microdissection of flash-frozen embryonic brain slices was performed to isolate the ventricular zone of neocortices, where neural progenitors reside (Fig EV1B). The concentrations of each member of the miR-34/449 family were determined by quantitative reverse transcription–polymerase chain reaction (qRT–PCR) and normalized to the concentration of miR-7-a-1, an abundant miRNA that regulates the p53 pathway in neural progenitors (Pollock *et al*, 2014). We found at least three members of the miR-34/449 family, miR449a, miR34a, and miR34b, expressed at levels similar to those of miR-7-a-1 (Fig 2A). *In situ* hybridization of E14 cortical slices further showed that miR-34b and miR449a are predominantly expressed in the ventricular and subventricular zone of the neocortex, where neural progenitors reside (Fig 2B and C). Thus, the abundance and expression pattern of miR-34 and miR-449 is consistent with a potential function in neural progenitors.

## Deletion of miR-34/449 perturbs cortical development

Genetic deletion of the miR-34/449 family in mice was previously shown to cause reduced brain size (Song *et al*, 2014) and perturbed development of intermediate forebrain structures (Wu *et al*, 2014), yet the specific developmental defects underlying these phenotypes were not determined. To further dissect the function of miR-34/449 during brain development, we generated double and triple knockout mice (DKO and TKO) of miR-449abc/34bc and miR-449abc/34a/34bc loci, respectively, crossing previously generated miR-449abc and miR-34a/34bc KO mice (Bao *et al*, 2012; Concepcion *et al*, 2012). This revealed that the reduced brain size of the miR-34/449 DKO/TKO mice (Fig 2D and E) was largely due to significantly thinner cortices, compared to heterozygote litter controls in young postnatal day 23 (P23) mice (Fig 2F and G). The brain size and cortical thickness phenotypes of miR-34/449 DKO and TKO mice were indistinguishable, indicating that miR-34a does not compensate for the deletion of other miRNA family members. For further loss-of-function phenotype characterization, we therefore combined DKO/TKO littermates into one group (subsequently referred to as KO). Together, these data show that miR-34/449 has an important function in the development of the mouse brain cortex.

## miR-34/449 regulates neurogenesis in the mouse cortex

Changes in cortical thickness observed at P23 in miR-34/449 KO mice might result from defective cortical neurogenesis during embryonic development. To test this, we first imaged brain sections of mouse embryos at embryonic day 16 (E16) stained for neuronal

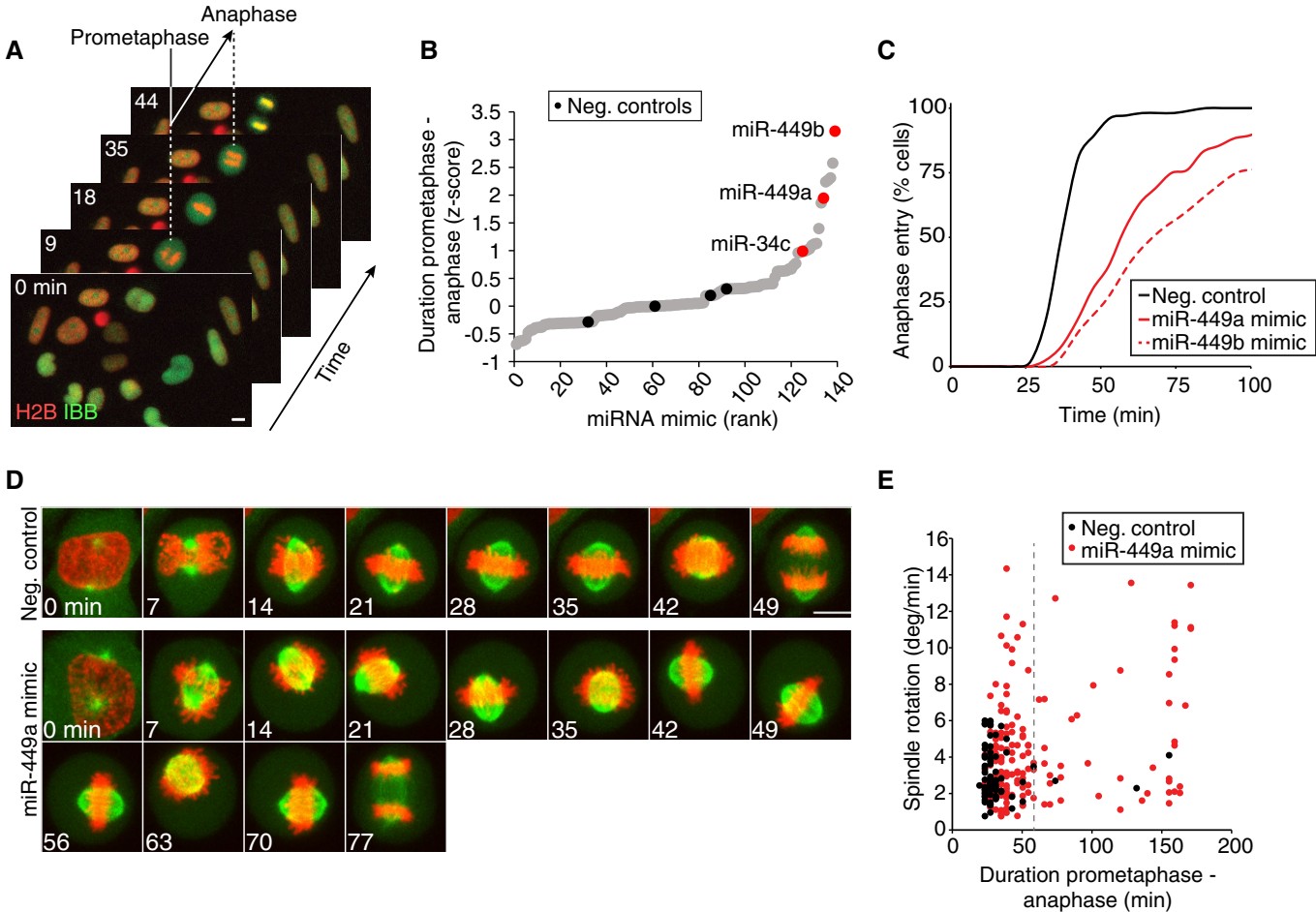

**Figure 1. Screen for candidate miRNAs involved in cell division.**

A   Live-cell imaging assay to detect mitotic perturbations. HeLa cells stably expressing a chromatin marker (H2B–mCherry; red) and a nuclear import substrate (IBB–eGFP; green) were imaged by automated live-cell microscopy and the duration of mitosis from prometaphase to anaphase onset was automatically determined for each dividing cell based on the time from nuclear envelope breakdown (mitotic entry) until nuclear reassembly (mitotic exit) (Schmitz *et al*, 2010). Scale bar, 10 μm.

B   miRNA mimic screen for miRNAs that regulate mitosis genes. miRNA mimics from a library of all embryonic cortically expressed miRNAs were transfected individually into HeLa cells and prometaphase to anaphase onset duration was determined as in (A). Individual points correspond to the mean z-score of the duration of prometaphase to anaphase onset determined in 2 independent experimental replicates for a given miRNA mimic.

C   Cumulative histograms of mitotic progression for control cells and for cells transfected with miRNA mimics. Nuclear envelope breakdown is at *t* = 0 min (*n* ≥ 71 in all conditions).

D   Confocal time-lapse microscopy images of HeLa cells stably expressing H2B–mCherry (red) and a microtubule marker (α-Tub–eGFP; green) 48 h after transfection of miR-449a mimic, or nontargeting control siRNA. Scale bar, 10 μm.

E   Quantification of spindle rotation and duration from prometaphase to anaphase onset in time-lapse movies as in (D). miR-449a mimic promotes spindle rotation. Duration from prometaphase to anaphase onset was defined as the time from nuclear envelope breakdown to anaphase onset. Spindle rotation during metaphase was measured as described in Materials and Methods. Individual data points correspond to single cells (*n* ≥ 72 in all conditions). Normality was tested with Kolmogorov–Smirnov test. Variance between samples was tested using *F*-test. Significance was tested by Welch's *t*-test: *P*-value = 2.598e-06 comparing spindle rotation of negative control vs. miR-449a mimic-transfected cells (all data points are compared). *P*-value = 0.0005621 for cells with prometaphase–anaphase onset duration lower than 60 min.

Source data are available online for this figure.

markers of the upper cortical layers, Satb2 (layers II–III) and Ctip2 (layer V). The number of Satb2+ cells, which are still migrating basally at this stage (Britanova *et al*, 2008) (Figs 3A and B, and EV3A), and Ctip2+ cells (Figs 3C and D, and EV3A) was significantly reduced in miR-34/449 KO (i.e., grouped DKO and TKO) mice, indicating a delay in late neuron generation during cortical development. We next imaged brain sections at E14 stained with the deep layer neuronal marker Tbr1 (layer VI). This showed that

the formation of the first layer during corticogenesis was also significantly impaired in miR-34/449 mice (Figs 3E and F, and EV3B). Altogether, these data show that miR-34/449 is important for the generation of several cortical layers at different stages during cortical neurogenesis.

It was previously shown that the observed reduction in cortical thickness in Dicer-ablated cortices was, in part, the result of the apoptosis of newborn neurons (De Pietri Tonelli *et al*, 2008). To

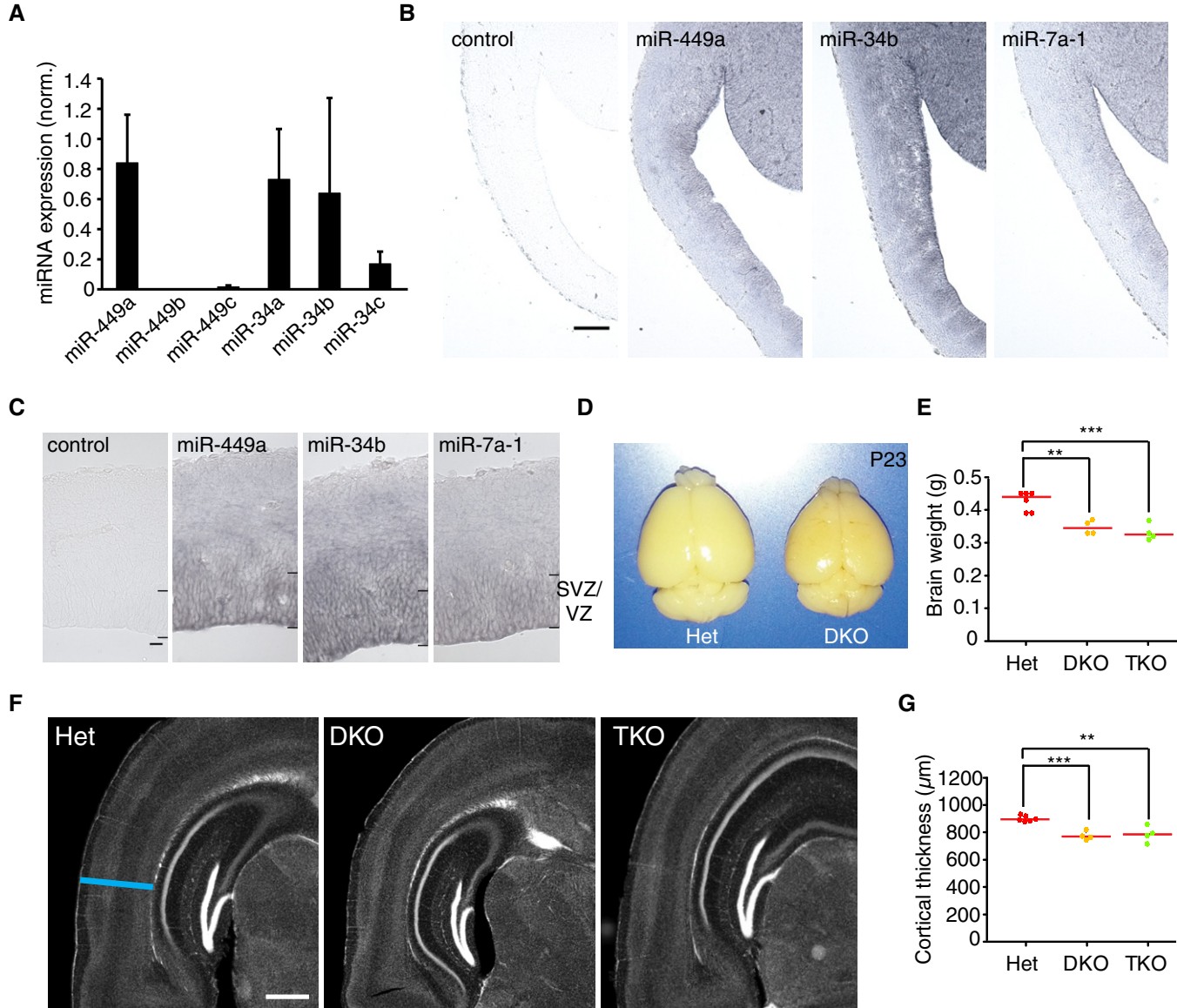

**Figure 2.   miR-34/449 family is expressed in neural progenitors and is required for normal cortex development.**

A     The expression levels of endogenous miR-34/449 family members were measured by RT–qPCR in ventricular zone samples derived by laser microdissection of mouse cortices at E14. The levels of the different miR-34/449 family members and miR-7a-1, a highly expressed miRNA relevant in cortical progenitor biology, were determined. All concentrations were normalized (norm.) using miR-7a-1 concentration (*n* = 8 cortices, 2 different litters). Error bars indicate standard error.

B, C   Expression analysis of miR-34/449 by *in situ* hybridization using locked nucleic acid (LNA) probes in wild-type cortices at E14. Mature miR-449, miR-34b, and miR-34c are preferentially expressed in the subventricular (SVZ) and ventricular (VZ) zones of the neocortex. Scale bar, 50 μm (B), 10 μm (C).

D, E   Brains of adult mice (P23) and quantification of brain weight. Dots indicate individual brains; red line indicates median. Mice lacking miR-449abc and miR-34bc (DKO) or miR-449abc, miR-34bc, and miR-34a (TKO) have significantly smaller brains compared to littermate controls (Het). Significance was tested by pairwise *t*-test with Bonferroni correction; Het (*n* = 6 brains, 2 different litters) vs. DKO (*n* = 4 brains, 2 different litters), \*\**P*-value = 0.00215; Het (*n* = 6 brains, 2 different litters) vs. TKO (*n* = 4 brains, 2 different litters), \*\*\**P*-value = 0.00054.

F, G   Confocal images of coronal brain sections (P23) and quantification of cortex width from miR-34/449 DKO and TKO mice and littermate controls (Het). Sections were stained with DAPI. DKO or TKO mice have significantly thinner cortices compared to littermate controls (Het) in adult mice (P23). Significance was tested by pairwise *t*-test with Bonferroni correction; Het (*n* = 6 brains, 2 different litters) vs. DKO (*n* = 4 brains, 2 different litters), \*\*\**P*-value = 0.00084; Het (*n* = 6 brains, 2 different liters) vs. TKO (*n* = 4 brains, 2 different litters), \*\**P*-value = 0.00211. Scale bar, 500 μm.

Source data are available online for this figure.

determine whether the neurogenic defects observed in miRNA-34/449 KO mice were the result of the apoptosis of differentiating neurons, we imaged brain sections of mouse embryos at E16 stained for TUNEL or activated caspase-3 (Fig EV2A and B). Cell death was not detected in both miR-34/449 KO mice and control littermates, suggesting that the deficiency in neuron generation in the miR-34/

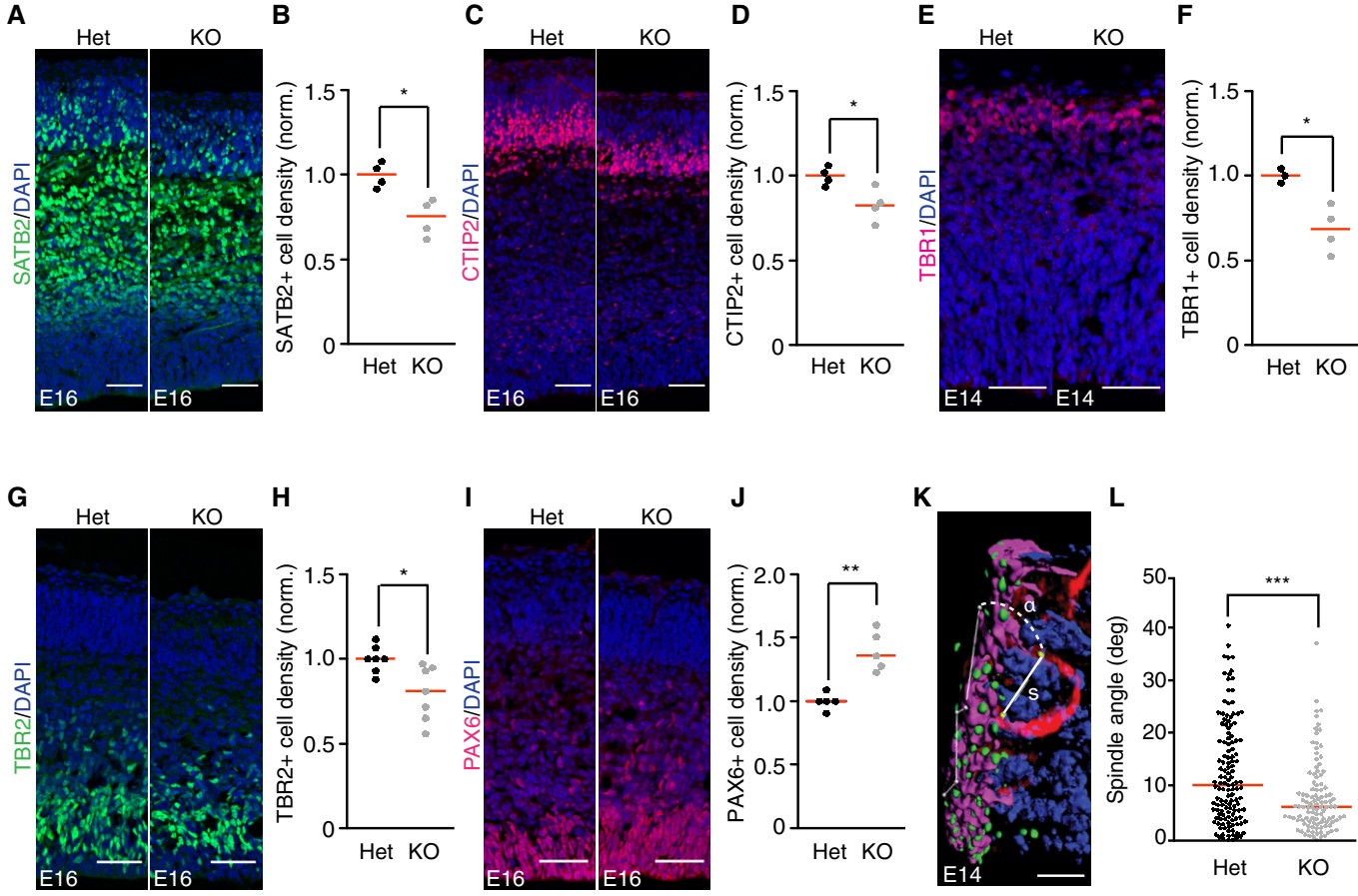

**Figure 3.   miR-34/449 family is required for timely cortical neurogenesis.**

A, B   Confocal images of coronal sections from E16 brains of miR-34/449 KO mice and littermate controls (Het), stained with anti-Satb2-antibody to label neurons of layers II–IV (green) and DAPI to label all cell nuclei (blue). Quantification of the number of Satb2-positive cells per 100 μm ventricular zone surface (*n* = 4 brains per genotype group, 2 independent litters). *P-value = 0.01146 (Het vs. KO).

C, D   Confocal images of coronal sections from E16 brains of miR-34/449 KO mice and littermate controls (Het), stained with anti-Ctip2 antibody to label V neurons and DAPI to label all cell nuclei (blue). Images were taken from same brain slices as shown in (A). Scale bars: 50 μm. Quantification of the number of Ctip2-positive cells per 100 μm ventricular zone surface (*n* = 4 brains per genotype group, 2 independent litters). *P-value = 0.03224 (Het vs. KO).

E, F   Confocal images of coronal sections from E14 brains of miR-34/449 KO mice and littermate controls (Het), stained with anti-Tbr1 antibody to label layer IV neurons and DAPI to label all cell nuclei (blue). Quantification of the number of Tbr1-positive cells per 100 μm ventricular zone surface (*n* = 3 and 4 brains per genotype group from 2 independent litters). *P-value = 0.01392 (Het vs. KO).

G, H   Confocal images of coronal sections from E16 brains of miR-34/449 KO mice and littermate controls (Het), stained with anti-Tbr2 antibody to label intermediate progenitors and DAPI to label all cell nuclei (blue). Quantification of the number of Tbr2-positive cells per 100 μm ventricular zone surface (*n* = 7 brains for each genotype group, 5 independent litters). *P-value = 0.01647 (Het vs. KO).

I, J   Confocal images of coronal sections from E16 brains of miR-34/449 KO mice and littermate controls (Het), stained with anti-Pax6 antibody to label radial glia progenitors and DAPI to label all cell nuclei (blue). Quantification of the number of Tbr2-positive cells per 100 μm ventricular zone surface (*n* = 5 brains per genotype group, 3 independent litters). **P-value = 0.002768 (Het vs. KO).

K   3D reconstruction of a dividing radial glial progenitor at early anaphase. A coronal section of an E14 brain of a heterozygous control mouse was stained with anti-phospho-vimentin antibody (red), anti-γ-tubulin antibody (green/yellow), phalloidin (magenta), and DAPI (blue) and imaged by 3D confocal microscopy. "s" indicates spindle axis, and "α" indicates angle relative to ventricular surface plane, which was determined by a vector path as indicated by thin white lines located on the left side of the image. Yellow dots highlight the centrosomes of the spindle poles from the analyzed dividing cell.

L   Quantification of spindle orientation in radial glial cells as in (K) for E14 brains of miR-34/449 KO and littermate controls (Het). Each dot represents a single dividing cell; ***P-value = 3.166e-05 (Het vs. KO) (*n* = 131 vs. 107 cells, *n* = 4 brains per genotype group, 2 independent litters).

Data information: Scale bars: 50 μm (A, C, E, G, I) and 5 μm (K). Bars indicate mean ± SEM. Data were normalized (norm.) to the ventricular zone surface analyzed (100 μm) and relativized to the heterozygous control average value. Statistical significance was tested by Welch's *t*-test.
Source data are available online for this figure.

449 KO is not the secondary effect of cell death/apoptosis during the differentiation process.

In some cell types, miR-34 expression is regulated by the p53 pathway that activates G1 arrest (Chivukula & Mendell, 2008) and thus might control the rates of cell cycle exit during neuronal development. To test whether the lower number of cortical neurons in miR-34/449 KO mice correlate with perturbed cell cycle progression, we injected day 15 pregnant female mice with 5-bromo-2′-deoxyuridine (BrdU) for

24 h, fixed E16 embryo brains, and performed double immunostaining for BrdU and the proliferation marker Ki67 (Fig EV2C and D). The abundance of BrdU$^+$ and Ki67$^+$ cells was indistinguishable in miR-34/449 KO and littermate heterozygous controls (Fig EV2E), suggesting that miR-34/449 is not relevant for cell cycle exit in neural progenitors. To further test whether the neurogenic defects in miR-34/449 mice correlate with perturbed S phase and/or overall cell cycle duration, we consecutively injected first (5-ethynyl-2′-deoxyuridine) EdU and 2.5 h later BrdU into day 15 pregnant female mice, fixed E16 embryo brains, and performed triple immunostaining for EdU, BrdU, and the progenitor markers Pax6 and Tbr2 (Fig EV2F and G). S phase and cell cycle duration was measured as in Martynoga *et al* (2005), revealing that neural progenitors divide once every 24 h during midneurogenesis (Noctor *et al*, 2004). The duration of S phase and overall cell cycle were indistinguishable between miR-34/449 KO and littermate heterozygous controls (Fig EV2H and I), indicating that miR-34/449 is not required for normal cell cycle progression of neural progenitors.

### Intermediate progenitor generation depends on miR-34/449

During cortical neurogenesis, intermediate progenitors delaminate from the ventricular to the subventricular zone of the cortex, where they subsequently divide symmetrically to generate two neurons (Lancaster & Knoblich, 2012; Williams & Fuchs, 2013). This pathway of indirect neurogenesis accounts for the majority of the neurons generated in the adult cortex (Sessa *et al*, 2008). Decreased rates of cortical neurogenesis have been associated with a reduced generation of Tbr2$^+$ intermediate progenitors (Postiglione *et al*, 2011). Imaging of E16 brain slices showed that the number of Tbr2$^+$ cells was significantly reduced in miR-34/449 KO mice (Figs 3G and H, and EV3C). Thus, the neurogenesis defect in miR-34/449 KO mice is associated with a reduced rate of intermediate progenitor generation.

Since intermediate progenitors were less abundant in miR-34/449 KO mouse cortices, we tested whether this was the consequence of an imbalance in the generation of radial glial cells. Imaging of E16 brain slices revealed that Pax6$^+$ radial glial progenitors were significantly more abundant in miR-34/449 KO mice (Figs 3I and J, and EV3D). Altogether, these data show that miR-34/449 is necessary for the correct balance of radial glial cell amplification and intermediate progenitor generation during cortical neurogenesis.

### miR-34/449 is necessary for proper mitotic spindle orientation radial glial cells

As spindle orientation is a major determinant of the fate of neural progenitors (Postiglione *et al*, 2011; Xie *et al*, 2013), we quantified the spatial organization of mitotic cells in the developing cortex. We stained E14 brain slices with a marker that selectively labels mitotic cells (phospho-vimentin), a spindle pole component (γ-tubulin), microtubules, and the mitotic cell surface (phalloidin) and recorded confocal image stacks. Semiautomated image analysis of 3D spindle orientation (Postiglione *et al*, 2011) in anaphase and early telophase cells revealed that radial glial cells in miR-34/449 KO mice had a significantly reduced angle relative to the ventricular surface compared with control heterozygous littermates (Fig 3K and L).

Thus, miR-34/449 is required for correct spindle orientation at the onset of cortical neurogenesis, which provides a possible explanation for the imbalance in the generation of intermediate progenitors and the reduced number of cortical neurons in miR-34/449 KO mice.

MiR-449 was previously shown to be required for vertebrate multiciliation by directly repressing the Notch pathway molecules Notch1 and Dll1 (Marcet *et al*, 2011). Since Notch activation correlates with radial glial progenitor identity maintenance (Gaiano *et al*, 2000) and inhibits neurogenesis (Louvi & Artavanis-Tsakonas, 2006; Kageyama *et al*, 2008), we tested whether the elevated number of radial glial progenitors might be the consequence of Notch1/Dll1 deregulation in the neocortex of miR-34/449 KO mice. qRT–PCR analysis of E14 mice cortices showed no significant difference in the mRNA levels of Notch1 and Dll1 (Fig 4A), suggesting that the increase in radial glial progenitors in miR-34/449 KO mice is not the result of increased Notch1/Dll1 mRNA levels.

### Spindle orientation factor JAM-A is regulated by miR-34/449 in the neocortex

The phenotypic analysis of developing brains suggests that miR-34/449 might contribute to cortical neurogenesis by regulating target genes relevant for spindle orientation. To search for candidate targets, we first performed genome-wide mRNA expression profiling of HeLa cells transfected with the miR-449a mimic. Among 395 downregulated mRNAs (Appendix Table S2), four had known functions in mitotic spindle regulation. To determine which of these candidate target genes might be regulated by miR-34/449 *in vivo*, we performed qRT–PCR of samples derived from E14 cortices of heterozygous controls and miR-34/449 KO mice. Only the junction adhesion molecule-A (JAM-A, also termed F11R) mRNA was significantly upregulated in cortices of miR-34/449 KO mice (Fig 4A), which was also consistent with JAM-A mRNA and protein repression by miR-449a mimic in HeLa and J110 cells (Fig 4B–D). Thus, miR-34/449 might contribute to correct spindle orientation during cortical neurogenesis, at least in part, through the regulation of JAM-A.

MiR-34/449 and JAM-A are both conserved throughout vertebrates (Marcet *et al*, 2011). Furthermore, mouse and human JAM-A contain conserved miRNA binding sites for miR-34/449 in their coding sequence (Fig EV4A and B). To test whether miR-34/449 directly targets mouse JAM-A, we generated vectors for expression of GFP–JAM-A fusion proteins, which contained either the wild-type miR-449a binding sites (GFP–JAM-A-WT) or several mismatch mutations in the miR-449a binding site (GFP–JAM-A-MUT) (Fig EV4C). Co-transfection of miR-449a mimic specifically reduced the protein levels of GFP–JAM-A-WT but not GFP–JAM-A-MUT (Fig 4E and F), indicating that JAM-A is a direct target of miR-34/449.

JAM-A is a member of the immunoglobulin superfamily, and it localizes to the intercellular contact sites of epithelial cells (Martin-Padura *et al*, 1998). Besides its well-characterized function in promoting intercellular junctions (Iden *et al*, 2012), JAM-A was shown to regulate planar spindle orientation during mitosis (Tuncay *et al*, 2015). To investigate whether JAM-A is expressed in the developing neocortex, we imaged brain sections of mouse embryos

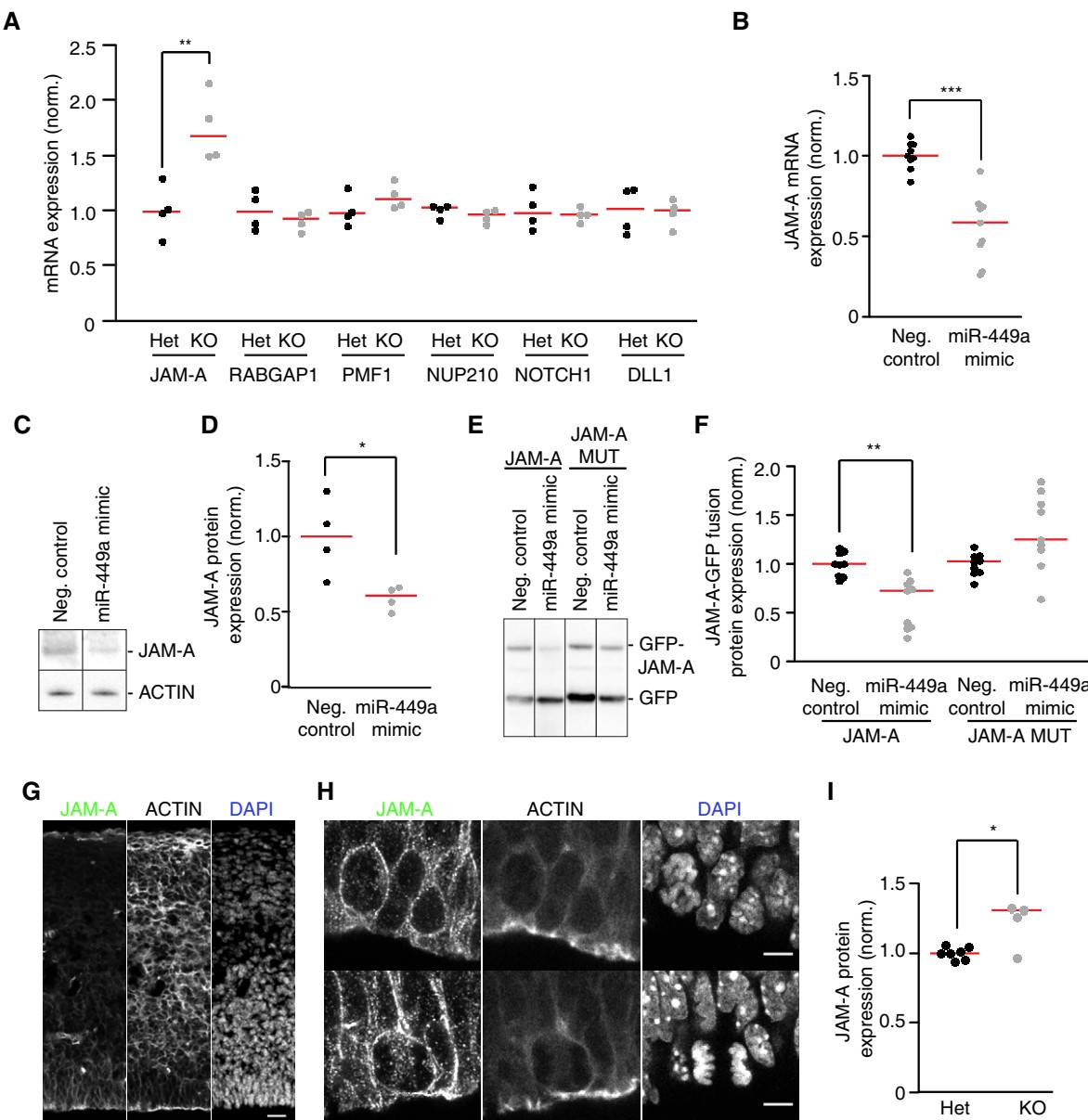

**Figure 4.  JAM-A is expressed in radial glial cells and is a direct target of miR-34/449.**

A    Quantification of mRNA expression by RT–qPCR of candidate miR-34/449 targets that might regulate spindle orientation in E14 cortex samples from miR-34/449 KO mice compared with littermate controls (Het). Expression was normalized (norm.) to phosphoglycerate kinase (PGK) mRNA levels. Significance was tested by Welch's t-test: **P = 0.009756 (JAM-A (Het) vs. JAM-A (KO)), all the rest Het vs. KO comparisons P = n.s. (n = 4 cortices per genotype group, 2 independent litters).

B    RT–qPCR quantification of JAM-A mRNA expression in HeLa cells 48 h after transfection of miR-449a mimic compared to cells transfected with nontargeting negative control siRNA. JAM-A mRNA expression was normalized (norm.) to glyceraldehyde-3-phosphate dehydrogenase (GAPDH) mRNA levels. ***P = 0.0001218 (Neg. control vs. miR-449a mimic) (n = 9 measurements from 3 independent replicates).

C, D    Mouse J110 cells were transfected with either miR-449a mimic or a negative control mimic. Whole-cell extracts were subjected to immunoblot analysis to detect endogenous JAM-A and actin. Actin was used to normalize JAM-A expression (norm.) as loading control. *P = 0.04454 (Neg. control vs. miR-449a mimic) (n = 4 independent wells, 2 independent transfections).

E, F    HeLa cells were cotransfected with either miR-449a mimic or a negative control mimic, together with JAM-A–GFP fusion protein and GFP. Whole-cell extracts were subjected to immunoblot analysis to detect GFP. GFP was used to normalize (norm.) by transfection efficiency as loading control. **P = 0.001176 (JAM-A WT Neg. control vs. miR-449a mimic) (n = 9 independent wells, 4 independent transfections).

G, H    Confocal images of coronal sections from E14 brains of wild-type mice stained with anti-JAM-A antibody (green), phalloidin to label actin (gray), and DAPI to label cell nuclei (blue). Scale bars: 10 μm (G), 5 μm (H).

I    Quantification of JAM-A protein expression in the ventricular zone of E14 brain coronal sections from miR-34/449 KO mice compared with littermate controls (Het) by immunofluorescence. Actin (phalloidin signal) was used to normalize (norm.) the expression levels of JAM-A. *P = 0.04842 (n = 5 (KO) or 7 (het) cortices per genotype group, 3 independent litters).

Data information: Red bars indicate the median. Statistical significance was tested by Welch's t-test.
Source data are available online for this figure.

    

at embryonic day 14 (E14) stained with antibody against JAM-A. This showed that JAM-A is predominantly expressed in radial glial cells of the subventricular and ventricular zones of the neocortex (Fig 4G and H). The similar expression patterns of JAM-A and miR-34/449a suggest that they might function in the same pathway. Furthermore, JAM-A localized to the cell cortex of dividing cells (Fig 4H), consistent with a potential contribution to spindle orientation. To determine whether JAM-A is upregulated in miR-34/449 KO vs. littermate heterozygous control mice, we quantified the JAM-A immunostaining in the ventricular zone of embryonic day 14 (E14) brain sections. JAM-A protein was significantly upregulated in cortices of miR-34/449 KO mice (Fig 4I), confirming the regulation of JAM-A by miR34/449 *in vivo*.

### JAM-A is required for normal spindle orientation

Given the established relevance of JAM-A for planar spindle orientation in Madin-Darby canine kidney cells (Tuncay *et al*, 2015), we next investigated whether it is required for normal spindle

orientation in HeLa cells. We thus used confocal time-lapse microscopy to image HeLa cells co-expressing a centrosome marker (centrin-2–eGFP) and a microtubule marker (α-tubulin fused to monomeric red fluorescent protein, α-tubulin–mRFP) (Logarinho *et al*, 2012) and quantified the mean spindle rotation velocity. RNAi depletion of JAM-A significantly increased spindle rotation and prolonged mitosis, similar to the phenotype observed after miR-449a mimic transfection (Fig 5A–C). In order to test whether JAM-A is indeed a functionally relevant target of miR-449 regarding spindle orientation, we investigated whether ectopically overexpressed JAM-A might suppress the miR-449 mimic-induced spindle rotation phenotype. We therefore generated a HeLa cell line stably overexpressing mouse JAM-A from a human elongation factor-1 alpha (EF1a) promoter, which we expected to maintain sufficient JAM-A protein levels even in the presence of miR-449a. Indeed, the cells stably overexpressing exogenous JAM-A maintained normal spindle orientation upon miR-449 mimic transfection, and the delay of mitotic progression was much less pronounced compared to wild-type cells (Figs 5D–F, and EV5D). Moreover, in a cell line

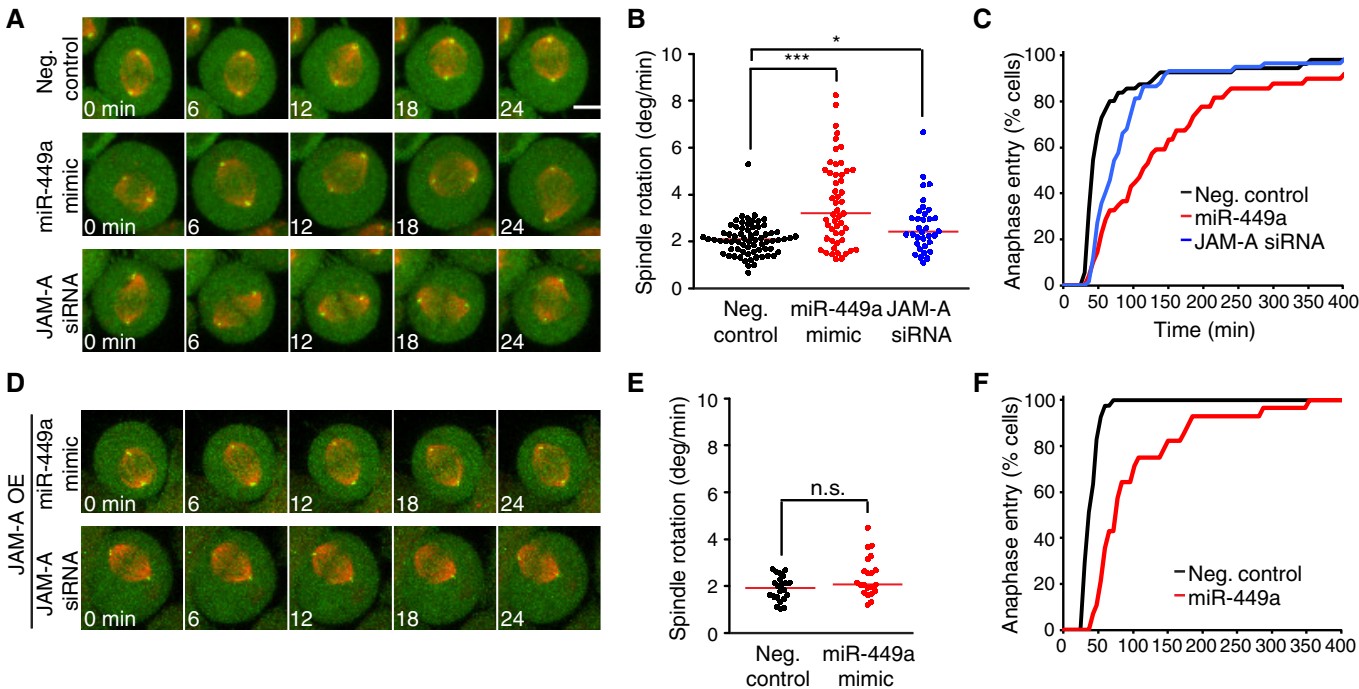

**Figure 5. JAM-A is required for normal spindle orientation and mediates miR-449-induced spindle defects.**

A  Confocal time-lapse images of metaphase HeLa cells stably expressing a centriole marker (centrin-2–eGFP; green) and a microtubule marker (α-tubulin–mRFP; red), 48 h after transfection of miR-449a mimic, JAM-A siRNA, or nontargeting negative control siRNA. Scale bar, 10 μm.

B  Quantification of spindle rotation for cells as shown in (A). Each dot represents a single cell. Red bar indicates median. Significance was tested by pairwise *t*-test with Bonferroni correction. ***P*-value = 2e-11 (Neg. control vs. miR449 mimic), *P*-value = 0.01928 (Neg. control vs. JAM-A siRNA).

C  Cumulative histograms of mitotic duration from nuclear envelope breakdown until anaphase onset, measured for cells as shown in (A); n ≥ 20 cells for all conditions, 2 independent replicates.

D  Confocal time-lapse images of metaphase HeLa cells stably expressing a centriole marker (centrin-2–eGFP; green), a microtubule marker (α-tubulin–mRFP; red), and mouse JAM-A, 48 h after transfection of miR-449a mimic, JAM-A siRNA, or nontargeting negative control siRNA. Scale bar, 10 μm.

E  Quantification of spindle rotation for cells as shown in (D). Each dot represents a single cell. Red bar indicates median. Significance was tested by pairwise *t*-test with Bonferroni correction. n.s., *P*-value = 0.17492 (Neg. control vs. miR449 mimic).

F  Cumulative histograms of mitotic duration from nuclear envelope breakdown until anaphase onset, measured for cells as shown in (D); n ≥ 20 cells for all conditions, 2 independent replicates.

Source data are available online for this figure.

stably overexpressing an unrelated protein, Cas9, miR-449 mimic transfection increased spindle rotation (Fig EV5A–D), validating the specific function of JAM-A in this phenotype rescue experiment. Altogether, these data are consistent with a role of JAM-A as part of a spindle orientation pathway regulated by miR-34/449 during cortical neurogenesis.

## Discussion

Our study reveals that the miR-34/449 family is required for proper spindle orientation in neural progenitors *in vivo*, which profoundly impacts mammalian neocortex formation. The miR-34 family was previously associated with negative regulation of cell cycle progression at the G1/S transition (Bommer *et al*, 2007; Chang *et al*, 2007; He *et al*, 2007; Tazawa *et al*, 2007), yet our data show that miR-34/449 deletion does not affect cell cycle duration in neural progenitors. Instead, miR-34/449 contributes to the production of intermediate progenitor cells during corticogenesis, likely through the regulation of mitotic spindle orientation in radial glial cells, promoting the generation of neurons by indirect neurogenesis. The miR-17-92 cluster was previously shown to inhibit the transition of radial glial cells to intermediate progenitors in the developing cortex (Bian *et al*, 2013; Nowakowski *et al*, 2013; Fei *et al*, 2014). Thus, multiple miRNA families contribute to cortical neurogenesis by balancing radial glial cell maintenance through different mechanisms.

Our data suggest that miR-34/449 regulates spindle orientation at least in part by directly targeting and inhibiting JAM-A. However, other target genes are likely involved given that RNAi of JAM-A yields a weaker phenotype than the miR-449a mimic and given that ectopic JAM-A overexpression does not fully compensate the miR-449a mimic-induced mitotic delay. Prior work showed that JAM-A controls spindle orientation and dynactin localization at the mitotic cell cortex via the activation of Cdc42 (Tuncay *et al*, 2015). Furthermore, Cdc42 is crucial for stable positioning of the metaphase spindle in the developing neuroepithelium of *Xenopus laevis* (Kieserman & Wallingford, 2009). These data and phenotypes revealed by our study suggest a spindle regulatory pathway that involves miR-34/449, JAM-A, and possibly Cdc42. This does not exclude the possibility, however, that the brain developmental defects observed in miR-34/449 KO mice might involve additional unknown targets of miR-34/449.

We have shown that miR-34/449 regulates spindle orientation in both neurons *in vivo* and epithelial cells (HeLa) in culture. Interestingly, miR-34/449 is also highly expressed in tracheal, fallopian and germinal epithelia (Song *et al*, 2014; Wu *et al*, 2014). An involvement of miR-449 in the development of tracheal epithelial tissues was previously attributed to misregulated differentiation of multiciliated cells (Marcet *et al*, 2011). Given that the organization of epithelia critically depends on mitotic spindle orientation (Ragkousi & Gibson, 2014), our study raises the possibility that miR-449 might have a more general function in tissue organization through spindle orientation. Interestingly, it was recently shown that spindle orientation is deregulated during the formation of the neural midline in a zebrafish Dicer mutant (Takacs & Giraldez, 2015), suggesting that miRNAs might have an important function in positioning the division plane of mitotic cells in several developmental contexts.

MiR-34/449 was previously shown to contribute to the development of other forebrain structures, including the putamen and the olfactory tubercle (Wu *et al*, 2014). The putamen is a major receiver of incoming axons from the cortex (Obernosterer *et al*, 2007). Connectivity between the cortex and the putamen requires the coordinated development of these structures for timely axogenesis from newborn cortical neurons to receptor spiny neurons (Goldman-Rakic, 1981). Thus, by coordinating distinct cell- and forebrain-specific programs, the miR-34/449 family emerges as a critical regulator of mammalian brain development.

## Materials and Methods

### Mice

Intercross mating mice miR-34a$^{-/-}$;miR-34bc$^{+/-}$;miR-449abc$^{-/-}$ and miR-34a$^{-/-}$;miR-34bc$^{-/-}$;miR-449abc$^{+/-}$ were established by a 2-step crossing of homozygous miR-449$^{-/-}$ (Bao *et al*, 2012) and miR-34a$^{-/-}$;miR-34bc$^{-/-}$ mice (Concepcion *et al*, 2012). miR-34a$^{-/-}$;miR-34bc$^{+/-}$;miR-449abc$^{-/-}$ and miR-34a$^{-/-}$;miR-34b~c$^{-/-}$;miR-449abc$^{+/-}$ intercross mice were crossed to generate miR-34a$^{-/-}$;miR-34bc$^{-/-}$;miR-449abc$^{-/-}$ (TKO) and miR-34a$^{+/+}$; miR-34b~c$^{-/-}$;miR-449abc$^{-/-}$ (DKO) mice as well as littermate heterozygous control (Het) mice. All mice were housed in a barrier animal facility at IMBA.

### Immunohistochemistry

Entire E14 mouse embryonic brains were dissected in ice-cold phosphate-buffered saline (PBS) and immediately fixed in 4% paraformaldehyde (PFA) overnight at 4°C, followed by cryoprotection in 30% sucrose until sunk to the bottom of Falcon tubes at 4°C. Brains were treated with a mix of 30% sucrose/O.C.T. (optimum cutting temperature) compound (Scigen Scientific) 1:1 overnight at 4°C. Brains were embedded in O.C.T.-filled cryomolds and freeze-dried on dry ice overnight before coronal sections were prepared using a Cryostat (Microm HM 550, Thermo Fisher Scientific). Coronal sections were postfixed in 1% PFA and permeabilized with 0.5% Triton X-100 in blocking solution (5% fetal bovine serum, 0.05% Triton X-100, and 5% BSA in PBS) for 10 min and then incubated with blocking solution for 30 min at room temperature, followed by the overnight incubation with primary antibody at 4°C. The following primary antibodies were used: mouse anti-γ-tubulin (1:1,000, Sigma), rabbit anti-Pax6 (1:250, Covance), rabbit anti-caspase-3 (1:250, Cell Signaling), chicken anti-Ctip2 (1:250, Abcam), rabbit anti-Tbr1 (1:250, Abcam), rabbit anti-Tbr2 (1:250, Abcam), mouse anti-Ki67 (1:100, Cell Signaling), and rabbit anti-JAM-A (aliquot 1168, Dr. Klaus Ebnet), as well as Alexa Fluor 647 phalloidin to stain actin (1:25, Life Technologies). For the quantitation of JAM-A protein *in vivo*, regions of interest (ROIs) were defined in the ventricular zone of embryonic stage 14 brain slices co-stained with JAM-A images and phalloidin. Mean JAM-A signal was normalized using mean phalloidin staining. Apoptosis was detected using the DeadEnd Fluorometric TUNEL assay (Promega) as described in the manufacturer's protocol. After incubation with the primary antibody, sections were washed in PBS, followed by incubation with appropriate fluorescence-conjugated secondary

antibodies (1:250, Life Technologies) for 1 h at 4°C. Sections were then treated with 2 µg/ml DAPI solution at room temperature for 15 min before mounting with mounting medium (DAKO).

## Laser capture microdissection

Entire E14 WT mouse embryonic brains were dissected in ice-cold PBS and immediately frozen in dry ice and stored in a deep freezer at −80°C overnight. Brains were sectioned into 15-µm-thick coronal slices using a cryostat (Microm HM 550, Thermo Fisher Scientific) at −15°C and mounted on PEN (polyethylene naphthalate) membrane slides. Sections were then fixed in ice-cold ethanol/acetone for 2 min. Cortical ventricular zone regions were laser-dissected with a LMD 6500 system (Leica Microsystems GmbH) and captured in an Eppendorf tube containing 200 µl of TRIzol reagent (Invitrogen). Ventricular zone dissections from at least 30 consecutive slices of the same brain were pooled together for RNA purification.

## Cortical thickness analysis

P23 mouse brains were dissected in ice-cold PBS and fixed overnight in 4% paraformaldehyde (PFA) at 4°C. Brains were washed in PBS before coronal sections were prepared using a Microtome (Leica VT 1000S). Coronal sections were postfixed in 4% PFA and permeabilized with 0.5% Triton X-100. Sections were washed in PBS, followed by incubation with 2 µg/ml 4′,6-diamidino-2′-phenylindole dihydrochloride (DAPI) solution at room temperature for 1 h before mounting with mounting medium (DAKO). Sections were imaged by fluorescence microscopy using Pannoramic 250 slide scanner and analyzed using Pannoramic Viewer 1.15 (3DHistech) and FIJI software (Schindelin *et al*, 2012).

## Live-cell imaging

Automated time-lapse microscopy for miRNA mimic screening was performed on a Molecular Devices ImageXpress Micro microscope equipped with a 10×, 0.5 N.A. S Fluor dry objective (Nikon) and laser-based autofocus. Cells were maintained in a microscope stage incubator at 37°C in a humidified atmosphere of 5% $CO_2$ throughout the experiment. 3D confocal microscopy was performed on a Zeiss LSM 780 microscope using a 40×, 1.3 N.A. oil DIC EC Plan-Neofluar objective (Zeiss). The microscope was equipped with piezo focus drives (Piezosystem Jena), custom-designed filters (Chroma), and an EMBL incubation chamber (European Molecular Biology Laboratory), which provided a humidified atmosphere at 37°C with 5% $CO_2$.

## Spindle angle analysis

E14 embryonic brain sections were stained with Alexa Fluor 647 phalloidin (1:50; Life Technologies) to detect the contour of the cells and immunostained with anti-phospho-vimentin (1:250; Cell Signaling) to identify dividing progenitors. Centrosomes were detected using anti-γ-tubulin (1:250; Sigma). Sections were imaged using a Zeiss LSM780 confocal microscope and 3D reconstructions from z-stack images were performed with IMARIS software (BITPLANE Scientific Software) as described previously (Postiglione *et al*, 2011). In short, *x*, *y*, and *z* coordinates of the two centrosomes of

anaphase or telophase radial glial cells, which divided adjacent to the ventricular surface, were annotated manually in 3D-rendered images. Five points embedded within the ventricular surface adjacent to the respective dividing progenitor were annotated to derive the best-fitting plane, which represents the ventricular surface by orthogonal distance regression. The angle between the vector connecting the centrosomes and the normal vector of the best-fitting plane for the ventricular surface was calculated using R scripts as described before (Postiglione *et al*, 2011).

## Analysis of time-lapse screening data

Automated image analysis was performed using the CellCognition software (Held *et al*, 2010) (http://www.cellcognition.org), based on a previously established processing pipeline to determine mitotic duration (Schmitz *et al*, 2010). Cell nuclei and mitotic chromosome regions were detected by local adaptive thresholding. Cytoplasmic regions were derived by region growing with a fixed size around the segmented chromatin regions. Next, texture and shape features were calculated for each object. Mitotic stages were defined by manual annotation, and annotated object sets were used to train a support vector machine classifier, based on radial kernel and probability estimates, for fully automated mitotic stage classification. Cells were tracked over time using a constrained nearest-neighbor approach, with an algorithm that supported trajectory splitting and merging. Mitotic events were detected in the graph structure on the basis of the transition from prophase to prometaphase. Nuclear envelope reassembly was defined as an increase in the ratio of the mean nuclear vs. cytoplasmic IBB–eGFP fluorescence > 1.5-fold above the ratio at the time of chromosome segregation. In the miRNA mimic screen, mitotic duration was normalized per 96-well plate to compensate for slight differences in the temporal sampling rate. *z*-scores were calculated based on the mean and standard deviation of all data points.

## Western blot analysis

Cells were seeded and transfected as described above, using LabTek II chambered coverslips (Thermo Scientific). Protein extracts were separated on 4–12% NuPAGE Bis-Tris gradient gels (Life Technologies) and transferred to PVDF (polyvinylidene difluoride) membranes (Amersham Hybond; GE Healthcare). Western blotting was performed by standard methods using antibodies against GFP (1:500, AB clone 2B6, MFPL Monoclonal Antibody Facility, Vienna), Cas9 (1:500, MFPL Monoclonal Antibody Facility, Vienna), JAM-A (1:100, AB H202-106 batch g3013, Santa Cruz Biotechnology), and actin (1:10,000, AB clone C4, Millipore). Western blotting imaging was performed using a ChemiDoc MP Imaging System (Bio-Rad). Western blot analysis was performed using Image Lab 5.1.1 (Bio-Rad) and FIJI software (Schindelin *et al*, 2012). GFP–JAM-A band intensities were normalized against corresponding GFP control in each lane for ratio calculation. JAM-A band intensities were normalized against corresponding actin control in each lane for ratio calculation.

## miRNA *in situ* hybridization

*In situ* hybridization was performed on frozen sections using locked nucleic acid (LNA) probes(Obernosterer *et al*, 2007). After

postfixation with 4% paraformaldehyde (PFA) for 10 min and acetylation with acetylation buffer for 10 min (1.33% triethanolamine, 0.25% acetic anhydride, 20 mM HCl), samples were treated with proteinase K for 5 min (10 mg/ml, IBI Scientific) and pre-hybridized (1× SSC, 50% formamide, 0.1 mg/ml salmon sperm DNA solution, 1× Denhart, 5 mM EDTA, pH 7.5) for 6 h at room temperature. Brain sections were hybridized with DIG-labeled LNA probes at RNA melting temperature (Tm) −30°C overnight (1:300 in hybridization buffer). The first wash was made at hybridization temperature for 15 min, after which 2 more subsequent washes were made at 4°C (1× SSC, 50% formamide, 0.1% Tween-20). After washing 2 times with pre-cooled 1× MABT, sections were blocked in blocking buffer (1× MABT, 2% blocking solution, 20% heat-inactivated sheep serum) for 2 h at RT and incubated with anti-DIG antibody (1:1,500, Roche) at 4°C overnight. Sections were washed 5 times for 20 min at RT with 1× MABT and 2 times for 10 min at RT with staining buffer (0.1 M NaCl, 50 mM MgCl$_2$, 0.1 M Tris–HCl, pH 9.5). Finally, sections were stained with 500 μl of BM purple, which was replaced every 6 h (Roche) at room temperature until ideal intensity was reached.

### Statistical analysis

All datasets were tested for normality using Kolmogorov–Smirnov test and for variance equality using the *F*-test. Depending on normality and variance equality, datasets were statistically compared using two-tailed Student's *t*-test or Welch's *t*-test. For some datasets containing multiple comparisons, we used ANOVA two-way test to measure statistical significance. Double blinding was applied for all experiments in Figs 3 and EV2.

**Expanded View** for this article is available online.

### Acknowledgements

The authors thank Andrea Ventura for providing miR-34 KO mice, Karin Aumayr, Tobias Mueller, Pawel Pasierbek, and Gabriele Petri of the IMBA/IMP BioOptics Core Facility for excellent support with microscopy, Bernhard Haubner, Marko Repic, Magdalena Renner, Mihaela Zeba, Jennifer Jurkin, Tanja Drexel, Michael Held, Michael H.A. Schmitz, Lisa Landskron, and Christoph Sommer for technical assistance, Stefan Schuechner and Egon Ogris of the Monoclonal Antibody Facility at MFPL—Vienna for the anti-GFP and anti-Cas9 antibodies, all members of the D.W.G. laboratory and Shan Bian for discussions, Javier Martinez and Luisa Cochella for discussions and comments on the manuscript, and Angela Anderson from Life Science Editors for editing assistance. The Gerlich laboratory has received financial support from the European Community's Seventh Framework Programme FP7/2007–2013 under grant agreement no 241548 (MitoSys) and no 258068 (Systems Microscopy), from an ERC Starting Grant (agreement no 281198), from the Wiener Wissenschafts-, Forschungs- und Technologiefonds (WWTF) (project nr. LS14-009), and from the Austrian Science Fund (FWF) (project nr. SFB F34-06). J.P.F. was supported by an EMBO long-term fellowship.

### Author contributions

JPF and DG conceived the study. JPF, CE, DG, and JAK designed experiments. JPF, CE, BM, and RS performed experiments. JPF, CE, BM, RS, DG, and JAK analyzed data. SY, HZ, KE, and WY provided reagents. JPF, JAK, and DG wrote the paper. DG supervised the project.

### Conflict of interest

The authors declare that they have no conflict of interest.

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
