## [Review Process File · The EMBO Journal]

Manuscript EMBO-2016-94056

MicroRNA-34/449 controls mitotic spindle orientation during mammalian cortex development

Juan Pablo Fededa, Christopher Esk, Beata Mierzwa, Rugile Stanyte, Shuiqiao Yuan, Huili Zheng, Klaus Ebner, Wei Yan, Juergen A. Knoblich, Daniel W. Gerlich

Corresponding authors: Juan Pablo Fededa and Daniel W. Gerlich, IMBA

Review timeline:

Submission date:	05 February 2016
Editorial Decision:	14 March 2016
Revision received:	23 June 2016
Editorial Decision:	12 July 2016
Revision received:	18 August 2016
Accepted:	06 September 2016

Editor: Anne Nielsen

Transaction Report:

1st Editorial Decision

14 March 2016

Thank you again for submitting your manuscript for consideration by the EMBO Journal and my apologies for the slightly extended duration of the review period in this case. Your study has now been seen by three referees and their comments are shown below.

As you will see from the reports, all referees express interest in the findings reported in your manuscript, although they differ somewhat in their assessments of the overall scope of the work. While ref #1 and #2 would recommend publication after significant revision, ref #3 is more hesitant and finds that causality in both miRNA regulation and the role for spindle orientation in neural progenitor fate specification would need to be supported further. However, you will also see that the first of three major points raised by ref #3 is largely recapitulated in the specific points requested by ref#1.

Given the referees' overall positive recommendation, I would therefore like to invite you to submit a revised version of the manuscript, addressing the comments of all three reviewers. I should add that it is EMBO Journal policy to allow only a single round of revision, and acceptance of your manuscript will therefore depend on the completeness of your responses in this revised version.

For the revised manuscript I would particularly ask you to focus your efforts on the following points:

- > Since both ref #1 and ref #3 raise concerns about the functional contribution from miRNA regulation here, I will ask you to address all points raised by ref #1, including the rescue experiments in vivo.
- > Please also address the comments made by ref #2
- > For ref #3, the first point largely overlaps with concerns from ref #1 and should be addressed, the second point can be discussed (although I would encourage you to include more data if it should be available)
- > Regarding the third major point of ref #3 this is a useful point that would strengthen the overall conclusion of the presented data and should be addressed if possible.

I realize that these are not trivial experiments to perform but in light of the input from the referees we will need the revised manuscript to include further data on miRNA-target causality in vivo. Given the work involved, I would understand if you should decide to rather seek rapid publication at a less demanding venue elsewhere. If you should choose to do so I would appreciate if you could let us know so that we can withdraw the manuscript from our system. I would also be happy to discuss the requirements for the revised manuscript further with you over the phone.

Thank you for the opportunity to consider your work for publication. I look forward to your revision.

REFeree COMMENTS

Referee #1:

In the present manuscript, Fededa et al. study the function of microRNAs in mitotic spindle orientation during mammalian cortex development. Using a miRNA mimic screen in HeLa cells, they identify members of the miR-34/449 cluster as important regulators of spindle orientation and cell division. The function of this cluster in vivo is validated by the use of miRNA triple knockout mice. Finally, JAM-A is proposed as an important direct target of miR-34/449 for spindle orientation and timely mitotic progression.

Overall, this is a well conceived study that describes a novel function of microRNAs in neuronal differentiation via the regulation of spindle orientation. The presented datasets are mostly convincing and adequately controlled. While the involvement of miR34/449 in cortical neurogenesis is well documented through the use of knockout mice, the significance of JAM-A regulation by miR34/449 for this process is less clear. Therefore, data supporting a direct interaction of miR34/449 and JAM-A during spindle orientation in cortical neurogenesis should be provided before this manuscript can be considered for publication in the EMBO Journal.

Specific comments:

1. In the screen, artificial miRNA gain-of-function using transfection of miRNA mimics was used to delineate candidate regulators of cell division. While this approach is useful in a screen setup, the function of endogenous candidate miRNAs that emerged from the screen should still be validated by the use of inhibitory antisense oligonucleotides or sponge transcripts.
2. The expression data on miR34/449 during cortical development could be strengthened by performing in situ hybridization analysis. This could also provide insight into the subcellular localization of these microRNAs.
3. The target validation for JAM-A is currently weak. In addition to mRNA data, the authors should provide protein data both from mice and cells. Moreover, miR34/449 inhibition should be used to demonstrate a negative regulatory role of the endogenous miRNAs on JAM-A mRNA and protein expression. Such experiments could be also done in the context of reporter gene constructs containing wild-type or mutant miR34/449 binding sites.
4. The presented data falls short of demonstrating a causal role of miR34/449 mediated JAM-A inhibition for normal spindle orientation and mitotic progression. Showing that JAM-A siRNA phenocopies miR-449a mimic is not sufficient to claim that they are in the same pathway. Therefore, the authors should attempt to rescue the miR-449 phenotype by restoring JAM-A expression, e.g. by

transfecting miRNA-resistant JAM-A expression constructs together with miR-449 mimic. Alternatively, a combination of JAM-A siRNAs and miR34/449 inhibitors could be used. Ultimately, it would be desirable to show that JAM-A upregulation in miRNA double/triple knockout mice is involved in cortex development, e.g. by electroporating JAM-A and control siRNAs.

Referee #2:

Fededa et al. demonstrated that microRNA-34/449 controls spindle orientation in radial glial cells during mouse cortex development. The authors identified miRNA-34/449 by using live-cell imaging-based high-content screening and identified its target, JAM-A, with genome-wide microarray analysis. The approaches are unique, and the individual data are of high quality and well documented. Since the contribution of microRNA to spindle orientation remains largely unknown, I believe that the findings presented in this study are of general interest to research pertaining to spindle orientation and neurogenesis.

Major points:

1. mRNA expression of JAM-A increased in miR-34/449 KO mice (Fig. 4A). To support this, it is important to analyze the localization of JAM-A in radial glial progenitors and to show the increase of JAM-A in miR-34/449 KO mice.
2. To support the authors' conclusion, it would be critical to analyze whether expression of JAM-A MUT (Fig. EV3) in radial glial progenitors causes a spindle misorientation phenotype similar to that in miR-34/449 KO mice (Fig. 3L).

Minor comments:

3. Please define what the symbol * in Table S1 indicates?
4. Appendix p16, line 21. Does "plates containing siRNA" mean "plates containing miRNA mimics"?

Referee #3:

Comments on "microRNA-34/449 controls mitotic spindle orientation during mammalian cortex development", by Fededa et al.

Fededa and colleagues performed a screen after transfection of 135 miRNA mimics in Hela cells and identified the miR-34 and miR-449 families as having an effect on cell division. Based on this, they then searched for possible effects of these miRNAs in the developing mouse brain by crossing existing knock-out mice known to have brain phenotypes. The authors confirm the KO mice have a reduced brain size compared to het. controls, and show that this could in part be due to a thinner cortex. They then go on to show that some populations of progenitors and neurons are altered in the KO. They propose that this is caused by changes in spindle orientation, due to angles slightly more aligned with the ventricular surface plane in mutants than in controls. While the results are intriguing and potentially interesting for the field, I find the data neither robust enough nor sufficient for publication, especially in an established broad interest journal. A revised manuscript may eventually be more suited to a specialized journal.

Major points:

- 1 Despite the robust wording used to imply or claim causal links between miR34 + miR449 perturbation, spindle misorientation, neurogenesis and JAM-A levels, the data supporting these claims are not sufficiently robust or controlled. No rescue or alternative experiments are presented to support that lack of miR34 and miR449 causes the mitotic, cell populations and tissue phenotypes presented.

2 The evidence for a universal causal link between spindle orientation and progenitor cell fate in the developing cortex is not as clear cut as the authors wish to suggest, and part of the relevant literature is present in some referenced reviews. Therefore, even if the authors were to confirm the role of these miRs in spindle orientation, this would not necessarily mean the orientation changes observed would be enough to explain the cellular and tissue changes. Other effects of the KO could be involved. Also, the misorientations measured are not very strong, and the statistical significance of the difference with controls may strongly depend on a few of the most angled orientations in the controls.

3 Most of the data presented was gained from only 2 independent samples per experiment. This seems low, especially in light of the relatively minor magnitude of some of the changes observed, either at spindle, cell populations or tissue level.

Minor points:

4 A good interpretation of the tissue data presented is difficult without showing the single channels for the images and only showing the merges. This is especially important in light of the low number of independent experiments performed.

5 The term "norm." in most figures is not described anywhere, which makes it difficult to interpret the graphs.

6 "Mitotic duration" was defined by the authors as the time between nuclear envelope breakdown and anaphase onset, but this is confusing and leaves out 3 of the 5 phases that contribute to the actual "mitotic duration". The authors should use a term that reflects the actual measurements they performed.

1st Revision - authors' response

23 June 2016

Response to the referee's comments (Fededa et al., EMBOJ-2016-94056)

We are pleased that all three referees find our study interesting and we are grateful for their constructive criticisms and suggestions how to improve the manuscript.

Referee #1 finds the study well-conceived and mostly adequately controlled. The referee requests additional experiments to determine the expression pattern of miR-34/449 by in situ hybridization, to validate JAM-A as a miR-34/449 target at the protein level, and to further test the relationship between miR-34/449-mediated regulation of JAM-A and the spindle orientation phenotype. We performed experiments to address these points, showing that miR-34/449 and JAM-A protein are co-expressed at E14 in the ventricular and sub-ventricular zone of the neocortex – the region where radial glial cells reside. We further show that JAM-A overexpression suppresses the spindle rotation phenotype caused by miR-449 mimic transfection, validating that JAM-A is a functionally relevant target of miR-449 in the control of spindle orientation.

Referee #2 finds the study interesting and the data of high quality. The referee requests additional experiments to determine the localization of JAM-A protein in the mouse cortex, as well as an experiment addressing functional relevance of miR-34/449-mediated JAM-A regulation for spindle orientation. To address the first point, we performed immunofluorescence staining, which showed that JAM-A is expressed in the subventricular and ventricular zone of the cortex, where radial glial cells reside and where miR-34/449 is expressed. We therefore used an in vitro rescue approach to further investigate functional relevance of JAM-A regulation by miR-34/449 for spindle orientation. We show that constitutive overexpression of JAM-A suppresses the miR-449 mimic-induced spindle rotation phenotypes in tissue culture cells, corroborating that JAM-A is a functionally relevant miR-449 target regarding the control of mitotic spindle orientation.

Referee #3 also finds the study interesting but raises concerns about sample sizes and statistical analysis. Moreover, the referee asks for a more careful discussion of the data, as brain tissue defects observed in miR-34/449 knockout mice might also involve perturbations independent of the

observed spindle misorientation phenotype. We have rephrased the manuscript accordingly and now better explain how the sample sizes and tests were designed. We also extended our study by a phenotype rescue experiment, which corroborates the proposed model of miR-34/449 regulating spindle orientation via JAM-A. Below, we provide detailed point-by-point response to all concerns raised by the referees.

Referee #1:

In the present manuscript, Fededa et al. study the function of microRNAs in mitotic spindle orientation during mammalian cortex development. Using a miRNA mimic screen in HeLa cells, they identify members of the miR-34/449 cluster as important regulators of spindle orientation and cell division. The function of this cluster in vivo is validated by the use of miRNA triple knockout mice. Finally, JAM-A is proposed as an important direct target of miR-34/449 for spindle orientation and timely mitotic progression.

Overall, this is a well-conceived study that describes a novel function of microRNAs in neuronal differentiation via the regulation of spindle orientation. The presented datasets are mostly convincing and adequately controlled. While the involvement of miR34/449 in cortical neurogenesis is well documented through the use of knockout mice, the significance of JAM-A regulation by miR34/449 for this process is less clear. Therefore, data supporting a direct interaction of miR34/449 and JAM-A during spindle orientation in cortical neurogenesis should be provided before this manuscript can be considered for publication in the EMBO Journal.

We are pleased that this referee finds our manuscript interesting and potentially suitable for publication in EMBO Journal. We agree with the referee that our study could be further strengthened by additional analysis of how miR34/449-mediated regulation of JAM-A might contribute to spindle orientation. To address this, we performed several new experiments. First, we show by in situ hybridization for miR-34b and miR-449a and immunofluorescence staining for JAM-A that these factors are co-expressed in the same brain regions at E14. Second, we quantified JAM-A protein levels in immunofluorescence stainings of brain slices and found that JAM-A protein is overexpressed in the ventricular and subventricular zone in miR34/449 knockout mice at E14. Third, we established phenotype rescue experiments, showing that overexpression of JAM-A suppresses the mitotic spindle orientation phenotype and mitotic delay caused by miR-449a mimic transfection in HeLa cells. These new experiments corroborate key conclusions of our study and we hope that the referee now finds it suitable for publication.

Specific comments:

1. In the screen, artificial miRNA gain-of-function using transfection of miRNA mimics was used to delineate candidate regulators of cell division. While this approach is useful in a screen setup, the function of endogenous candidate miRNAs that emerged from the screen should still be validated by the use of inhibitory antisense oligonucleotides or sponge transcripts.

We agree with the referee that it would be interesting to compare gain-of-function phenotypes with loss-of-function phenotypes also in vitro. However, by RNA sequencing we found that all members of the miR-449/34 family are expressed at extremely low levels in HeLa cells (data not shown, miR-449a: 8 counts per million reads (CPM), miR-449b: 2 CPM, miR-34c: 0.5 CPM). We also screened a panel of 10 other cell lines (HEK293, Rpe1, Hc11, Mcf10A, J110, T47D, BXD1425-EPN, Skbr3, Huh7, HeLa Kyoto) for the expression of two fully conserved representative members of the miR-449abc and miR-34bc loci, miR-34c and miR-449a. These miRNAs were not expressed at high levels in any of the tested cell lines (all measurements were below 100 CPM). That this miRNA family is not expressed in the tested tissue culture cells is not unexpected, given its very specific expression pattern in vivo (e.g., in cortical progenitors or multiciliated cells of the trachea). The low expression in tissue culture cells was in fact the main reason for us to set up and perform all miRNA loss-of-function experiments in vivo.

2. The expression data on miR34/449 during cortical development could be strengthened by performing in situ hybridization analysis. This could also provide insight into the subcellular localization of these microRNAs.

We thank the referee for suggesting these additional validation experiments. We performed in situ hybridization experiments for two miR34/449 family members, which showed expression predominantly in the ventricular and sub-ventricular zone of the mouse neocortex at E14 – the region where radial glial cells reside (new Fig. 2B-C). The spatial resolution and signal-to-noise levels in the images, however, was not sufficient to determine the subcellular localization of these miRNAs. Nevertheless, we think that the new in situ hybridization data address the main concern about the expression in the relevant brain regions at E14.

3. The target validation for JAM-A is currently weak. In addition to mRNA data, the authors should provide protein data both from mice and cells. Moreover, miR34/449 inhibition should be used to demonstrate a negative regulatory role of the endogenous miRNAs on JAM-A mRNA and protein expression. Such experiments could be also done in the context of reporter gene constructs containing wild-type or mutant miR34/449 binding sites.

We thank the referee for suggesting how to improve the target-validation. To assess the regulation of JAM-A at the protein level, we tested by Western blotting several commercial and proprietary antibodies for specific detection of JAM-A. We found only one antibody (SC-59845 anti-mouse JAM-A) to specifically detect JAM-A protein – yet only the mouse but not the human ortholog. We therefore used a J110 mouse cell line to investigate the effect of miR-449a mimic transfection on JAM-A protein levels. Western blotting showed that miR-449a mimic transfection significantly reduced endogenous JAM-A protein levels (new Fig. 4C-D), thus corroborating that JAM-A is a target of miR449a.

To address the regulation of JAM-A protein levels in mouse brains, we performed immunofluorescence staining in E14 mouse cortices. We found JAM-A predominantly expressed in the ventricular and sub-ventricular zone of the neocortex, where radial glial cells reside (new Fig. 4G-H). Importantly, JAM-A protein levels in the ventricular and sub-ventricular zone were significantly higher in E14 brains of miR-34/449 KO mice (new Fig. 4I), providing further evidence that miR-34/449 regulates JAM-A protein expression in radial glial cells.

4. The presented data falls short of demonstrating a causal role of miR34/449 mediated JAM-A inhibition for normal spindle orientation and mitotic progression. Showing that JAM-A siRNA phenocopies miR-449a mimic is not sufficient to claim that they are in the same pathway. Therefore, the authors should attempt to rescue the miR-449 phenotype by restoring JAM-A expression, e.g. by transfecting miRNA-resistant JAM-A expression constructs together with miR-449 mimic. Alternatively, a combination of JAM-A siRNAs and miR34/449 inhibitors could be used. Ultimately, it would be desirable to show that JAM-A upregulation in miRNA double/triple knockout mice is involved in cortex development, e.g. by electroporating JAM-A and control siRNAs.

We thank the referee for suggesting how to further test a potential causal relationship between miR34/449-mediated JAM-A inhibition and normal spindle orientation and mitotic progression. To establish phenotype rescue assays, we generated a HeLa cell line stably expressing exogenous wildtype JAM-A from a constitutive Human elongation factor-1 alpha (EF1a) promoter. We expected sufficiently high protein expression levels from this promoter to sustain JAM-A protein even in the presence of miR-449 mimics. Indeed, the cells stably overexpressing exogenous JAM-A maintained normal spindle orientation upon miR-449 mimic transfection, and the delay of mitotic progression was very much reduced compared to wildtype cells (revised Fig. 5). This provides strong evidence that miR-449 and JAM-A function in the same pathway regulating mitotic spindle orientation.

We also attempted to establish phenotype rescue assays in mouse embryos. To manipulate the levels of JAM-A in radial glial cells at E14, we followed two different strategies. We injected into the ventricle of E11/12 embryos (miR-34/449 KO) either lentivirus expressing CAS9/gRNA to induce frame-shift loss-of-function mutations, or shRNA targeting JAM-A mRNA. These experiments turned out to be technically very challenging, however, and we could not achieve sufficiently high infection rates at conditions that were still compatible with embryonic survival up to E14 (both in the control and KO background).

We think that the new in vitro phenotype rescue data in combination with the new in vivo co-expression data (as discussed above) provide strong support for the main points of our

manuscript and hope that the referee now finds our manuscript suitable for publication in EMBO Journal.

Referee #2:

Fededa et al. demonstrated that microRNA-34/449 controls spindle orientation in radial glial cells during mouse cortex development. The authors identified miRNA-34/449 by using live-cell imaging-based high-content screening and identified its target, JAM-A, with genome-wide microarray analysis. The approaches are unique, and the individual data are of high quality and well documented. Since the contribution of microRNA to spindle orientation remains largely unknown, I believe that the findings presented in this study are of general interest to research pertaining to spindle orientation and neurogenesis.

We are pleased that this referee finds our work of general interest and high quality and we thank for the constructive suggestions how to improve our manuscript.

Major points:

1. mRNA expression of JAM-A increased in miR-34/449 KO mice (Fig. 4A). To support this, it is important to analyze the localization of JAM-A in radial glial progenitors and to show the increase of JAM-A in miR-34/449 KO mice.

We agree with the referee that investigating JAM-A protein localization and abundance in the developing brain is an important experiment. Following the suggestion, we performed immunofluorescence staining for JAM-A in E14 mouse cortical slices and found JAM-A predominantly expressed in the ventricular and sub-ventricular zone of the neocortex, where radial glial cells reside (new Fig. 4G, H). Furthermore, JAM-A protein levels in the ventricular and sub-ventricular zone were significantly higher in E14 brains of miR-34/449 KO mice (new Fig. 4I). Together, these new data corroborate that miR-34/449 regulates JAM-A protein expression in radial glial cells.

2. To support the authors' conclusion, it would be critical to analyze whether expression of JAM-A MUT (Fig. EV3) in radial glial progenitors causes a spindle misorientation phenotype similar to that in miR-34/449 KO mice (Fig. 3L).

We thank the referee for suggesting how to further test the causal relationship between miR-34/449-mediated JAM-A regulation and the spindle misorientation phenotype. We agree that investigating the role of JAM-A in vivo in radial glial cells would be a good approach, and we have attempted to implement embryonic infection experiments using lentiviral shRNA expression constructs against JAM-A. Unfortunately, we faced technical difficulties in these experiments, as transfected embryos were aborted before they reached E14.

We therefore sought for alternative ways how to further test the functional relevance of miR-34/449-mediated JAM-A regulation regarding spindle orientation. Towards this aim, we established phenotype rescue experiments in tissue culture cells. To maintain high levels of JAM-A protein after transfection of miR-449 mimics, we generated a HeLa cell line stably overexpressing JAM-A from a constitutive Human elongation factor-1 alpha (EF1a) promoter. Even though the exogenous JAM-A is still sensitive to miR-449 regulation, we expected the residual JAM-A protein expression from this strong promoter to be sufficient to suppress the spindle orientation phenotype. Indeed, miR-449 mimic transfection did not cause excessive spindle rotation in the cells expressing exogenous JAM-A, in contrast to wildtype HeLa cells transfected with miR-449 mimic, which exposed a strong spindle rotation phenotype (revised Fig. 5A-D). This provides strong evidence that JAM-A is a functionally relevant target of miR-449 in regulating spindle orientation in HeLa cells.

In the revised manuscript, we further show that both JAM-A protein (new immunofluorescence data, Fig. 4G-I) and several miR-34/449 family members (new in situ hybridization data, Fig. 2B-C) are both expressed in the subventricular and ventricular zone at E14. The co-expression data in mouse brains and the in vitro phenotype rescue data provide strong support for

functional relevance of miR-34/449-mediated regulation JAM-A regarding proper spindle orientation during brain development.

Minor comments:

3. Please define what the symbol * in Table S1 indicates?

We added a definition of "" symbol in the beginning of the Table S1: "*" represents miRNA product from the 3' arm of the hairpin"*

4. Appendix p16, line 21. Does "plates containing siRNA" mean "plates containing miRNA mimics"?

Yes – we thank the referee for pointing this out. We corrected the definition of "plates containing siRNA" to "plates containing miRNA mimics in the Appendix.

Referee #3:

Comments on "microRNA-34/449 controls mitotic spindle orientation during mammalian cortex development", by Fededa et al.

Fededa and colleagues performed a screen after transfection of 135 miRNA mimics in HeLa cells and identified the miR-34 and miR-449 families as having an effect on cell division. Based on this, they then searched for possible effects of these miRNAs in the developing mouse brain by crossing existing knock-out mice known to have brain phenotypes. The authors confirm the KO mice have a reduced brain size compared to het. controls, and show that this could in part be due to a thinner cortex. They then go on to show that some populations of progenitors and neurons are altered in the KO. They propose that this is caused by changes in spindle orientation, due to angles slightly more aligned with the ventricular surface plane in mutants than in controls. While the results are intriguing and potentially interesting for the field, I find the data neither robust enough nor sufficient for publication, especially in an established broad interest journal. A revised manuscript may eventually be more suited to a specialized journal.

The referee finds our results intriguing and potentially interesting, but raises concerns regarding statistical analysis and some of the data interpretations. We are grateful for the constructive criticisms, which helped us to improve the revised manuscript as explained below.

Major points:

1 Despite the robust wording used to imply or claim causal links between miR34 + miR449 perturbation, spindle misorientation, neurogenesis and JAM-A levels, the data supporting these claims are not sufficiently robust or controlled. No rescue or alternative experiments are presented to support that lack of miR34 and miR449 causes the mitotic, cell populations and tissue phenotypes presented.

We agree that phenotype rescue experiments are a good approach to further test a causal relationship between miR-34/449-mediated JAM-A regulation and the observed phenotypes in cells and tissues.

We therefore established phenotype rescue experiments in tissue culture cells to test the relationship between miR-34/449-mediated downregulation of JAM-A and spindle orientation. To maintain high levels of JAM-A protein after transfection of miR-449 mimics, we generated a HeLa cell line stably overexpressing JAM-A from a constitutive Human elongation factor-1 alpha (EF1a) promoter. Even though the exogenous JAM-A is still sensitive to miR-449 regulation, we thought that the residual JAM-A protein expression from this strong promoter might be sufficient to suppress the spindle orientation phenotype. Indeed, miR-449 mimic transfection did not cause excessive spindle rotation in the cells expressing exogenous JAM-A, in

contrast to wildtype HeLa cells transfected with miR-449 mimic, which exposed a strong spindle rotation phenotype (revised Fig. 5A-D). This provides evidence that JAM-A is a functionally relevant target of miR-449 regarding correct spindle orientation in HeLa cells.

We also attempted to manipulate JAM-A expression levels in mouse embryos to perform similar phenotype rescue experiments in brain tissues. We used lentiviral infection to induce CRISPR/Cas9-mediated frameshift-knockout, or to express shRNA targeting JAM-A in brains of E14 embryos. Unfortunately, neither approach yielded sufficient infection efficiencies to score the very rare cell division events, which precluded us to draw any conclusions from these experiments. Given that these experiments require complicated crossings, it appears unlikely that the technical problems can be solved within reasonable time.

The in vitro phenotype rescue data together with the co-expression data of JAM-A and miR-34/449 in E14 mouse brains provide additional support for a model of miR-34/449-mediated regulation JAM-A for proper spindle orientation during brain development. We addressed the lack of in vivo rescue experiments by mentioning in the manuscript the possibility of potential other target genes involved in the developmental defects:

*“Our data suggest that miR-34/449 regulates spindle orientation at least in part by directly targeting and inhibiting JAM-A. However, other target genes are likely involved given that RNAi of JAM-A yields a weaker phenotype than the miR-449a mimic, and given that ectopic JAM-A overexpression does not fully compensate the miR-449a mimic-induced mitotic delay. Prior work showed that JAM-A controls spindle orientation and dynactin localization at the mitotic cell cortex via the activation of Cdc42 (Tuncay et al, 2015). Furthermore, Cdc42 is crucial for stable positioning of the metaphase spindle in the developing neuroepithelium of *Xenopus laevis* (Kieserman & Wallingford, 2009). These data and phenotypes revealed by our study suggest a spindle regulatory pathway that involves miR-34/449, JAM-A, and possibly Cdc42. This does not exclude the possibility, however, that the brain developmental defects observed in miR-34/449 KO mice might involve additional unknown targets of miR-34/449.”*

Regarding the concern about the robustness of the data, we would like to point out that all of our conclusions are based on significance by statistical testing, using sample numbers similar to many other studies published in this field. We provide the detailed explanations on statistical testing and sample numbers in the figure legends and methods section.

2 The evidence for a universal causal link between spindle orientation and progenitor cell fate in the developing cortex is not as clear cut as the authors wish to suggest, and part of the relevant literature is present in some referenced reviews. Therefore, even if the authors were to confirm the role of these miRs in spindle orientation, this would not necessarily mean the orientation changes observed would be enough to explain the cellular and tissue changes. Other effects of the KO could be involved. Also, the misorientations measured are not very strong, and the statistical significance of the difference with controls may strongly depend on a few of the most angled orientations in the controls.

The orientation of neural progenitor spindles is widely considered to be a key parameter influencing the lineage specification in the developing mouse brain. Given that miR-34/449 knockout leads to a spindle orientation phenotype in radial glia cells as well as a lineage specification phenotype that was previously described to result from similar spindle misorientations, we think that the model presented in our manuscript is the most conceivable interpretation. However, the referee correctly points out that it is theoretically possible that brain developmental defects might arise through misregulation of other unknown targets of miR-34/449, potentially through mechanisms other than spindle misorientation. We now mention this possibility in the revised manuscript discussion section: “This does not exclude the possibility, however, that the brain developmental defects observed in miR-34/449 KO mice might involve additional unknown targets of miR-34/449.”

The referee is concerned about the statistical significance of the spindle misorientation phenotype and asks if the differences might be explained by outlier data points. We would like to note that we are aware of this potential problem and therefore chose a non-parametrical statistical test, which is robust towards outliers (Mann-Whitney U test). While we agree that the effect strength is not very high in this experiment, we would like to note that it is within the range reported for other published spindle misorientation phenotypes.

3 Most of the data presented was gained from only 2 independent samples per experiment. This seems low, especially in light of the relatively minor magnitude of some of the changes observed, either at spindle, cell populations or tissue level.

This comment might be based on a confusion of the sample number of animals versus the number of independent litters. We would like to clarify that we used at least 4 animals for each genotype group in all experiments (except for the analysis of the Tbr1 marker with 3 animals for each genotype group). Given the complicated crossings required to generate the multi-gene knockouts, the sample numbers used in our study follow the standards in this field. Importantly, statistical testing validates that all of the conclusions drawn on the used sample numbers are significant.

Minor points:

4 A good interpretation of the tissue data presented is difficult without showing the single channels for the images and only showing the merges. This is especially important in light of the low number of independent experiments performed.

We agree that single channel representations improve the clarity of our figures and thus included a new supplementary Fig. EV4 containing all single channel images of the tissue immunofluorescence stainings. We also replaced merged channel images in Fig. EV2 to single channel images to improve the clarity of the results.

5 The term "norm." in most figures is not described anywhere, which makes it difficult to interpret the graphs.

We now define the term "norm." in the figure legends of the revised manuscript.

6 "Mitotic duration" was defined by the authors as the time between nuclear envelope breakdown and anaphase onset, but this is confusing and leaves out 3 of the 5 phases that contribute to the actual "mitotic duration". The authors should use a term that reflects the actual measurements they performed.

We thank the referee for the suggestion how to clarify the description of mitotic timing measurements. We now use the term: "Duration prometaphase – anaphase"

References

Kieserman EK, Wallingford JB (2009) In vivo imaging reveals a role for Cdc42 in spindle positioning and planar orientation of cell divisions during vertebrate neural tube closure. Journal of cell science 122: 2481-2490

Tuncay H, Brinkmann BF, Steinbacher T, Schurmann A, Gerke V, Iden S, Ebnet K (2015) JAM-A regulates cortical dynein localization through Cdc42 to control planar spindle orientation during mitosis. Nature communications 6: 8128

2nd Editorial Decision

12 July 2016

Thank you for submitting the revised version of your manuscript. It has now been seen by two of the original referees whose comments are shown below.

As you will see the referees are overall impressed by the experiments included in the revised manuscript and appreciate the difficulties you have encountered with the in vivo rescue. However,

while ref #2 is consequently fully supportive of publication, ref #1 asks for the inclusion of an additional control in the HeLa cell rescue experiment.

In light of the comments from our referees I would invite you to submit a final revision in which you include this cell-based rescue experiment using an unrelated protein as a control. In addition, referee #1 has a few lingering concerns with the statistical analysis that I would ask you to comment on.

Thank you again for giving us the chance to consider your manuscript for The EMBO Journal, I look forward to your revision.

REFEREE COMMENTS

Referee #1:

By following most of the referees' suggestions, the authors have substantially improved the manuscript.

However, in my opinion, one of my main concerns was not adequately addressed. Namely, the claim that JAM-A is an important downstream target of miR-449a in mitotic spindle orientation is not fully supported by the presented data. In Fig. 5 of the revised manuscript, the authors now present results from overexpression of JAM-A in the context of miR-449a mimic transfection ("rescue" condition). However, in order to make the claim that JAM-A expression rescues the miR-449a mimic phenotype, the authors have to compare miR-449 mimic (including overexpression of an unrelated protein, which was omitted) to miR-449 mimic + JAM-A overexpression. Is this significant?

Also, it was a little bit disturbing that the authors simply lumped together the original and new datasets, although the conditions were not identical (JAM-A siRNA was only conducted in the first set of experiments). In my opinion, it would be more appropriate to show the two datasets independently and make the respective statistical assessment (two-way ANOVA).

Referee #2:

For this revised manuscript, the authors have performed additional experiments to address the comments of the reviewers. Although it is unfortunate that embryonic infection experiments did not work due to technical difficulties, alternative in vitro rescue approaches support their conclusion. I believe the manuscript is ready for publication.

2nd Revision - authors' response

18 August 2016

Response to the referee comments (Fededa et al., EMBOJ-2016-94056)

We are pleased that both referees appreciate the improvements of our revised manuscript. While referee #2 finds all concerns satisfactorily addressed and supports publication without further changes, referee #1 requests one additional control experiment to further validate the specificity of the in vitro rescue of the miR-449 phenotype by overexpressed JAM-A. Referee #1 further suggests a re-layout of Figure 5 to display data from separate experiments in distinct panels. These are good suggestions to further improve our manuscript and we have addressed them as proposed by the reviewer. We now show that overexpression of an unrelated protein that is not target of miR-449 (Cas9) does not rescue the spindle orientation phenotype induced by miR-449 mimic. We also addressed the statistical concerns by re-layouting Figure 5 as suggested.

Referee #1:

By following most of the referees' suggestions, the authors have substantially improved the manuscript.

However, in my opinion, one of my main concerns was not adequately addressed. Namely, the claim that JAM-A is an important downstream target of miR-449a in mitotic spindle orientation is not fully supported by the presented data. In Fig. 5 of the revised manuscript, the authors now present results from overexpression of JAM-A in the context of miR-449a mimic transfection ("rescue" condition). However, in order to make the claim that JAM-A expression rescues the miR-449a mimic phenotype, the authors have to compare miR-449 mimic (including overexpression of an unrelated protein, which was omitted) to miR-449 mimic + JAM-A overexpression. Is this significant?

The referee is concerned that the phenotype rescue resulting from JAM-A overexpression might be an unspecific bystander effect of protein overexpression in general, rather than a specific effect of JAM-A. We think that this is unlikely, given that the cell line used for this assay already overexpressed two marker proteins (centrin-2-GFP and α -tubulin-RFP). Nevertheless, we agree that the suggested additional negative control by overexpression of an unrelated protein could further strengthen this experiment.

We therefore generated a new cell line stably overexpressing the unrelated bacterial protein Cas9 (but no gRNA), using the same lentiviral backbone vector as used for JAM-A overexpression. Overexpressed Cas9 did not rescue the spindle rotation phenotype induced by miR-449 mimic (new Fig. EV5), validating that the rescue presented in the previous version of our manuscript was specifically caused by JAM-A.

Based on two-way ANOVA testing (with Bonferroni correction) the differences between (miR-449 mimic + JAM-A overexpression) versus (miR-449 mimic + Cas9 overexpression) were significant ($p = 4.8e-09$). This test also revealed significant differences comparing presence or absence of miR-449 mimic in the original cell line only expressing the fluorescent marker proteins ($p = 2e-11$), but no significant differences comparing presence or absence of miR-449 mimic in the JAM-A-overexpressing cells ($p = 0.17492$).

Also, it was a little bit disturbing that the authors simply lumped together the original and new datasets, although the conditions were not identical (JAM-A siRNA was only conducted in the first set of experiments). In my opinion, it would be more appropriate to show the two datasets independently and make the respective statistical assessment (two-way ANOVA).

We had pooled only samples from identical experimental conditions in our previous manuscript. We do agree, however, that it might be better to present data from different cell lines in separate plots and have revised Fig. 5 accordingly. We also present the new control experiment in a separate panel (EV5).

Referee #2:

For this revised manuscript, the authors have performed additional experiments to address the comments of the reviewers. Although it is unfortunate that embryonic infection experiments did not work due to technical difficulties, alternative in vitro rescue approaches support their conclusion. I believe the manuscript is ready for publication.

We are pleased that referee #2 finds our manuscript ready for publication and appreciate again the constructive criticisms and suggestions raised during review process.

Corresponding Author Name: JUAN PABLO FEDEDA

Journal Submitted to: EMBO JOURNAL

Manuscript Number: EMBOJ-2016-94056